# Repellency Mechanism of Natural Guar Gum-Based Film Incorporated with Citral against Brown Planthopper, *Nilaparvata lugens* (Stål) (Hemiptera: Delphacidae)

**DOI:** 10.3390/ijms23020758

**Published:** 2022-01-11

**Authors:** Xiubing Gao, Xianfeng Hu, Feixu Mo, Yi Ding, Ming Li, Rongyu Li

**Affiliations:** 1Institute of Plant Protection, College of Agriculture, Guizhou University, Guiyang 550025, China; gxb527@163.com (X.G.); huxianfenggzu@163.com (X.H.); gzmfx@sina.com (F.M.); 18768689597@163.com (Y.D.); 2Guizhou Tea Research Institute, Guizhou Province Academy of Agricultural Science, Xiaohe District, Guiyang 550006, China; 3Guizhou Key Laboratory for Agricultural Pest Management in Mountainous Region, Guizhou University, Guiyang 550025, China

**Keywords:** *Nilaparvata lugens*, guar gum film, citral, repellent effect, RNA sequencing

## Abstract

Using of plant essential oil that coevolved as a defense mechanism against agriculture insects is an alternative means of controlling many insect pests. In order to repel brown planthoppers (BPHs), the most notorious rice insect pest, a new film based on guar gum incorporated with citral (GC film) was formulated, which was effective while being environmentally friendly. In this paper, the effect and mechanism of GC film repellency against BPHs were determined. Repellent activity test and olfactory reaction analysis showed that GC film had repellency effect against BPHs, with repellency of 60.00% and 73.93%, respectively. The result of olfactory reaction indicated that GC film repellency against BPHs relied on smell. EPG analysis showed the proportion and mean duration of np waveform were significantly higher than in CK and increased following the treatment concentration, which indicated that GC film affected the recognition of BPHs to rice. Further analysis by RNA sequencing analysis showed a total of 679 genes were significantly upregulated and 284 genes were significantly downregulated in the BPHs fed on the rice sprayed with GC film compared to control. Odorant-binding protein (OBP) *gene 797* and gustatory receptor gene (GR)/odorant receptor (OR) *gene 13110* showed a significant decrease in differential expression and significant increase in differential expression, respectively. There were 0.66 and 2.55 differential expression multiples between treated BPHs and control, respectively. According to the results described above, we reasoned that GC film repellency against BPHs due to smell, by release of citral, caused the recognition difficulties for BPHs to rice, and *OBP gene 797* and *GR/OR gene 13110* appeared to be the crucial candidate genes for GC film repellency against BPHs. The present study depicted a clear and consistent repellency effect for GC film against BPHs and preliminarily clarified the mechanism of GC film as a repellent against BPHs, which might offer an alternative approach for control of BPHs in the near future. Our results could also help in the development and improvement of GC films.

## 1. Introduction

The brown planthopper (BPH, *Nilaparvata lugens* Stål, Hemiptera: Delphacidae) is an oligophagous insect that feeds only on wild rice (*Oryza rufipogon* Griff.) and cultivated rice *(Oryza sativa* L.), primarily on cultivated rice, which is one of the world’s most important food crops and a staple food for over half of the population in Asian countries. Serious destructive acts, high fecundity and a remarkable migration make it one of the most notorious insect pests of rice.

The use of chemical insecticides is still the main strategy for BPH management. However, the shortcomings of cost in terms of labor, cost, and environment are becoming more prominent as the BPH has evolved different levels of resistance to many major classes of insecticides [1,2,3,4,5], and there are concerns about the human health and environment consequences of insecticide residue [6,7]. Therefore, the search for alternative and environmentally friendly strategies to control BPH has been an increasing interest. One strategy relies on the use of plant secondary metabolites that have coevolved as a defense mechanism against agriculture insects [8,9].

Citral, one typical kind of plant secondary metabolite and major compound of essential oils (Eos), displays excellent antimicrobial activity against pathogens [10,11] and certain insecticidal activity against pest [11,12,13,14]. Because of being susceptible to oxidative degradation and instable in water under neutral pH, citral easily loses biological activity under normal conditions. Thus, improving the stability of citral could be a key to enhance the efficacy of this plant secondary metabolite [14]. According to this purpose, we developed a film that was based on the excellent film-forming matter guar gum, and incorporated the bioactive essential oil citral, which repelled BPHs effectively and was environment friendly [15]. In the current study, we examined the repellency effect and mechanism of GC film against BPH. According to previous studies, Eos and other secondary metabolites, as well as their mixtures, have shown moderate repellency activities against many insects [14,16,17,18,19,20,21,22,23]. Therefore, the GC film contains citral, also speculated to have this effect, as the major component.

Based on the description above, the researchers were interested to test the effect of different GC film-forming emulsion concentrations on the number of the BPHs that landed on rice seedlings. After the confirmation of effectiveness, the feeding behavior of BPHs after being fed GC film-treated rice seedlings was assessed by electrical penetration graph (EPG) technique. Furthermore, the impact on BPH transcriptome was also assessed by high-throughput RNA sequencing technology. The results from this study could provide valuable information on the mechanisms of plant secondary metabolites against insects and increase our understanding of the effect of GC film against BPH.

## 2. Results

### 2.1. The Repellency Effect of GC Film to BPH

In the behavioral assay of BPHs that selected and landed at different treated rice seedlings, it showed that female adults were more inclined to choose rice seedlings sprayed with distilled water, for the number of BPHs landed on GC film-treated rice seedlings was lower than at control (Figure 1a). The repellency was increased with the treated concentration of GC film-forming emulsion, and was significantly high when was treated with the highest concentration. At the highest concentration (100 dilution multiple of GC film-forming emulsion), 60.00% repellency was recorded (Figure 1b). Bioassays in a four-arm olfactometer revealed that female adult BPHs did not prefer a high concentration of GC film-forming emulsion, for the number of BPHs at the 100 and 200 dilution multiples concentrations were significantly lower than at water and air (Figure 2a). Likewise, the olfactory reaction analysis showed a similar pattern to that of the feeding choice test, for the repellency also increased with the treated concentration of GC film-forming emulsion, and 73.93% repellency was recorded at the highest concentration (Figure 2b). The results of both the feeding choice test and the olfactory reaction analysis clearly showed that GC film has a repellency effect against BPHs, and olfactory reaction analysis indicated GC film repellency against BPHs relies on smell.

### 2.2. EPG Analysis

Five distinct waveforms of 6 h EPG recordings of BPH feeding behavior were identified, including np, ph, N4-a, N4-b, and N5 (Figure 3), coinciding with the EPG waveform characteristics of BPH aspiration behavior as described by Seo et al. [24] and He et al. [25]. The comparison waveforms of EPG response variables of BPHs feeding on rice plants treated with different dilution multiples of film-forming emulsions are shown in Figure 4. The results showed that in a 6 h recording period, there were significant differences in some waveforms between BPHs that fed on plants treated with GC film and those that fed on the CK. The proportion and mean duration of np waveform were significantly higher than for the CK. Further, following the increase in treatment concentration, the proportion and mean duration of np waveform were increased. At the highest concentration (100×), 37.25 ± 3.43% and 66.32 ± 8.89 min of proportion and mean duration were recorded, 6.69 and 4.13 times that of the CK, respectively (Figure 4a,b). The numbers of ph, N4-a, N4-b, and N5 waveforms were not significantly different or showed no regularity (Figure 4c). These results showed that the np waveform was significantly influenced when treated with GC film, for the duration of np waveform in EPG recordings was significantly prolonged. These results further confirmed that GC film had a repellency effect against BPH and repellency against BPH relies on smell.

### 2.3. Transcriptome Analysis

#### 2.3.1. Illumina Sequencing and Sequence Assembly

As can be seen from Table 1, a total of 15,0302,587 high-quality sequences were obtained after sequencing quality control, and the percentage of Q30 bases in each sample was not less than 93.55%. The total number of high-quality sequences obtained from the samples was relatively high, and the proportion of reference genes obtained by alignment was more than 75.70%. The proportion of sequences aligned to a single position and the proportion of sequences aligned to multiple positions were more than 66.10% and 8.22%, respectively. The proportion of sequence alignment to positive chain and negative chain was above 36.92% and 36.49%, respectively. The results show that the sequencing quality is good, and the high-quality sequence number obtained can be used for subsequent biological analysis.

#### 2.3.2. Differentially Expressed Genes Analysis

The results of differentially expressed genes in BPHs for 48 h are shown in Figure 5 and Figure 6. Compared with the control group, the expression levels of 17,116 genes of BPHs in treated group were different. Set selection significantly differentially expressed genes of conditions for |log2 Fold change| 0.585 or higher and q < 0.01 [a Fold change gene expression said multiples, q (padj value) is correct after the *p* value], a total of 963 differentially expression genes exist significant differences. Among them, 679 genes were significantly up-regulated and 284 genes were significantly down-regulated, as shown in Figure 5.

#### 2.3.3. Functional Enrichment Analysis

To understand the functional distribution of differentially expressed genes, Gene Ontology (GO) and Kyoto Encyclopedia of Genes and Genomes (KEGG) enrichment pathways analyses was performed. The results indicated that 21, 13, and 18 genes were classified into biological function, molecular function, and cellular components, respectively, by GO enrichment pathways analysis (Figure 7). The classification results showed that there was a greater distribution of catalytic activity (3540 genes), binding (3533 genes), and transporter activity (481 genes) differential genes, and overall, 968 gene expression profiles were in more detailed subcategories related to localization.

To search for metabolic or signal transduction pathways that were significantly enriched, KEGG pathway enrichment analysis was performed (Figure 8). The top eight KEGG pathways that significantly over-presented were purine metabolism (23), longevity-regulating pathway—multiple species (18), biosynthesis of amino acids (15), protein processing in endoplasmic reticulum (13), spliceosome (12), peroxisome (11), MAPK signaling pathway—fly (10), and carbon metabolism (10). The DNA replication (6) was significantly down-regulated, which are coordinately involved in the repellent response of BPHs.

#### 2.3.4. Repellency-Related Genes Analysis and qRT-PCR Validation

We focused on the chemosensory genes as repellency-related genes, for the previous research results indicated that GC film repellency against BPHs relies on smell, by EPG analysis. Through NCBI BLAST searching, we found that two chemosensory genes, *gene 797* and *gene 13110*, showed significant differential expression in the transcriptome data. The *gene*
*797*, which belongs to odorant-binding protein genes (OBPs) and is related to the function of odorant binding, showed a significant decrease in differential expression (0.66 differential expression multiple in treated BPHs compared to control). The *gene 13110*, which belongs to gustatory receptor genes (GRs)/odorant receptor genes (ORs) and is related to the gustatory and odorant receptor, was significant upregulated in its expression (2.55 differential expression multiple in treated BPHs compared to control, statistically significant at *p* < 0.05) (Table 2). To check whether the change trend of these two genes is consistent with that of previous results, qRT-PCR was performed (Table 3). The differential expression multiples for the two genes were 0.75 and 2.60, respectively, exactly consistent with previous results, showing that the transcriptome data we obtained were credible (Figure 9). According to these results and the results described earlier, we reasoned that GC film repellency against BPHs relies on smell, by release of citral, which causes the recognition difficulties for BPHs to rice. Moreover, *OBP gene 797* and *GR/OR*
*gene 13110* appeared to be the crucial candidate genes for GC film repellency against BPHs (Figure 10).

## 3. Discussion

Plant essential oils are considered to be an alternative means to synthetic compounds for controlling many insect pests [26,27,28]. One of most widely recognized effects shown by plant essential oils is a repellency activity on insect pests, and this could cause a significant reduction of the target pest populations on plants. Citral, one typical kind of plant secondary metabolite and a major compound of essential oils (EOs), also exhibits repellent effects on some insect pests [29], such as *Aedes aegyptis* [12], *Aedes albopictus* [13,30], *Rhopalosiphum padi* [31], *Megalurothrips sjostedti* [32], and *Bactrocera cucurbitae* [33]. In the present study, GC film, which incorporated the bioactive essential oil citral, showed a repellency effect against BPH, as determined by the 60.00% repellency recorded.

Many essential oils have shown repellency against biting arthropods [34,35,36] and against agricultural pests [37,38,39]. Zhang et al. tested 21 essential oils and 17 showed significant repellency on yellow jackets (mainly *Vespula pensylvanica* (Saussure)) and paper wasps (mainly *Polistes dominulus* (Christ)), and E/Z-citral, among 11 active compounds, showed a significant repellency on vespid workers [16]. Similar results were also observed in *Tanacetum parthenium* and *Tanacetum vulgare*. Lazarević et al. reported that at higher EO concentrations, both third instar larvae and adult *T. parthenium* and *T. vulgare* required less time to choose an untreated leaf disc. Additionally, EO provoked rapid movement away from the treated leaf disc especially at the highest concentration [40].

Our findings regarding film repellency are very interesting, as little is known about how repellents act on this target insect. To figure this out, EPG analysis was used to interpret the repellent act. The EPG technique is mostly used to study the real-time probing and ingestion behavior of sucking insects, for an insect’s stylet-probing behavior can be recorded by digital signal and afterwards transformed into waveform and then analysis by bundled software. EPG was first applied to research on the feeding behavior of aphids, then widely used in research on the feeding behavior of BPHs [13,24,41,42,43,44,45,46,47,48,49].

In the present study, we identified five typical waveforms for the feeding behavior of BPHs, which quietly coincided with those previously described [24,25]. In the 6 h recording period of our study, BPHs that fed on GC film-treated rice spent 2.20–3.32 min to ascertain whether to feed, which was significantly longer than that of the control, with 0.88 min. Further, the proportion and mean duration of np waveform increased with the increase of GC film treatment concentration, while ph, N4-a, N4-b, and N5 waveforms were not significantly different or showed no regularity. The np waveform, which is a straight line in EPG waveform and corresponds to nonpenetration, indicates that no feeding occurs, or that the stylet has not been inserted into the plant [44]. Nonpenetration is one of the BPHs continuous feeding behavior before probing. In this action, BPH searches for probing sites and ascertains whether to probe. In essence, it is a recognition process for BPH to rice. GC film significantly increased the duration of np waveform, indicating that GC film significantly influenced the recognition of BPH to rice.

To further investigate the mechanisms of *GC film* repellent against BPH, we carried out transcriptome analysis. Using Illumina HiSeq 2000 platform, a total of 15,0302,587 high-quality sequences were obtained after sequencing quality control, and the percentage of Q30 bases in each sample, which corresponds to a base accuracy of 99.9%, was not less than 93.55%. These results demonstrated that our transcriptome quality was high enough for further mining of the olfactory genes of BPHs. Through NCBI BLAST searching, we found two chemosensory genes, odorant binding protein gene (OBP) and gustatory receptor gene (GR)/odorant receptor gene (OR), significantly decreased in differential expression and significantly increased in differential expression, respectively. To further confirm this, we used qRT-PCR to check, and the results showed that the transcriptome data we obtained were credible. The behaviors of BPHs were triggered by volatiles and extracts from rice, and chemosensation, especially olfaction, plays a key role in locating host plants. Therefore, we reasoned that OBP and GR/OR appeared to be the crucial candidate genes for GC film repellency against BPHs. The number of chemosensory genes in our study using transcriptome were quite few. One reason might be that chemosensory genes are low-abundance genes in insects. Another might be that BPHs are monophagous, feeding only on rice plants, thus their low chemosensory genes numbers appear adequate to adapt to limited hosts compared to those of other omnivorous insect species [50], such as *Acyrthosiphon pisum* [51], *Tribolium castaneum* [52], and *Anopheles gambiae* [53]. Third might that the repellency of GC film against BPH is mainly through citral as the odorant, which only evoked specific responses of the olfactory system.

OBPs are small (15 kDa), soluble proteins, composed of six α-helices, six cysteine residues, and three disulfide bridges [54], and are very concentrated in the lymph of chemosensory sensilla [55]. The OBPs of BPH have been characterized and found to participate in olfaction, with some teepees and ketones as potential attractants [43,56,57]. He et al. reported *NlugOBP3*, which identified in BPHs as a facilitator of rice plant seeking and involved in nymph olfaction on rice seedlings, had markedly higher binding ability and wider binding spectrum than did the other two OBPs, *NlugOBP1* and *NlugOBP2* (He et al., 2011). Zhou et al. identified 10 OBPs, including seven previously unidentified, and reported four OBPs appeared to be antenna-specific because they were highly and differentially expressed in male and female antennae [58]. The expression changes of OBPs might be connected with improving detection sensitivity of the olfactory system and the function to connect external odorants to olfactory receptor neurons. These results were consistent with those of the olfactory reaction analysis. However, which, or which combination, are the target genes for GC film repellency against BPHs in our study remains unclear and requires further study.

The insect GR family originates from GR-like genes from some of the earliest animals, such as the cnidarian *Nematostella vectensis* and *Acropora millepora* [59,60]. Insect GR genes are usually considered as sugar/bitter receptors and mediate food preference [61,62,63], as well as being involved in other physiological functions such as sensing light and heat [64]. The insect OR family evolved from the GR family, which has been indicated in some original insect species, and most ORs have not been deorphanized in single insect species [65]. The ORs comprise is one of the insect odorant receptors families and appear to be the major determinant of the odor response [66]. With respect to differential expression of transcriptome data in our study, the OBPs were significantly decreased while GRs/ORs were significantly increased. The qRT-PCR results confirmed the observed downregulation of OBPs and the upregulation of GRs/ORs. This means that the GC film, with mainly citral as the odorant, repels against BPHs by inhibiting the expression of OBP genes, while simultaneously promoting the expression of GR/OR genes. To elucidate the detailed mechanism, further study is needed.

## 4. Materials and Methods

### 4.1. Plants and Insects

The three-line hybrid rice plant, “Jinyou 785”, was used for laboratory tests. BPHs were originally obtained from a rice paddy in Jiuzhou town, Huangping county, Guizhou Province, China (27.00 N, 107.74 E) and reared continuously via rice [15].

### 4.2. Repellent Activity Test

#### 4.2.1. Feeding Choice Test

The feeding choice of BPHs to differently treated rice seedlings was tested by the food-in-cage method as described by Kim et al. [67] with slight modifications. Rice seedlings were prepared by spraying four concentrations of each film-forming emulsion (100, 200, 400, and 800 dilution multiples) of GC film, as described by Gao et al. [15], whose method description is partly reproduced. Then, the rice seedlings were presented in a cage (25 cm × 20 cm × 15 cm, wrapped with 200 mesh polyester netting), and spraying with distilled water was set as the control. Fifteen one-day-old female adult BPHs were introduced into the cage after 4 h of starvation (but with access to water). The number of BPHs that selected and landed at different treated rice seedlings was respectively counted and recorded after 4 h under laboratory conditions (27 °C ± 1 °C, 16:8 light/dark photoperiod, and 70–80% relative humidity). Every test was performed in triplicate, and BPH repellency was calculated by a following equation: Repellency (%) = (B − A)/(A + B) × 100, where A was the number of BPHs that landed on treated rice seedlings and B was the number of BPHs that landed on controlled rice seedlings, respectively.

#### 4.2.2. Olfactory Reaction Analysis

The olfactory reaction analysis of BPHs to GC film was carried out according to the method stated by Sai et al. (2006) [68] and Mao et al. (2018) [69], with minor adjustments. Four concentrations of GC film emulsion (100, 200, 400, and 800 dilutions) were prepared, with water and fresh air as control. One milliliter of each solution was absorbed by pipette, respectively, and liberated on cotton to make it fully absorbed, then the two kinds of cotton were placed at intervals in 4 flavor bottles of four-arm olfactometer (R = 15 cm, 3 cm high, and 8 cm total thickness, YM4-30-300, Shanghai, China). After that, a total of 50 one-day-old female adult BPHs were selected after 4 h of starvation, but with access to water, and introduced into the central activity room of olfactory instrument after adapting to the environment for 30 min. To avoid the influence of light on the behavior of the BPHs, darkness was kept throughout the experiment. The airflow velocity was 0.8 L/min, and the test environment conditions were the same as the above. After 4 h, the number of BPHs within 18 cm in the control area and the treatment area was counted. Each treatment concentration was repeated 3 times. After each test, the test unit was cleaned with anhydrous ethanol several times in preparation for the next round of tests. Repellency (%) = [(C − T)/(C + T)] × 100%, where C is the number of BPHs in the control solution area, and T is the number of BPHs in the treatment solution area.

### 4.3. EPG Analysis

The feeding behavior of BTHs on GC film-treated rice was recorded and classified using direct current (DC) electrical penetration graph (EPG) (Giga-8; EPG Systems, Wageningen, The Netherlands), as described by Zhao et al. [70]. Rice seedlings grown in a greenhouse under natural light for 2 weeks after sowing were transplanted to 1000 mL plastic cups (20 cm diameter × 15 cm height, each cup contained 1 seedling) and grown for another 3 weeks. The well-grown rice in plastic cup was chosen and sprayed with four concentrations of each film-forming emulsion (100, 200, 400, and 800 dilution multiples) of GC film, with the treatment of spraying distilled water as control. One-day-old female adults BPHs that had been provided water only (water-saturated cotton) for 2 h were connected to a biological current amplifier by gold wire electrodes (2–3 cm long and 20.0 μm in diameter), then introduced into the plastic cup and carefully placed onto the stem of the rice. The digital signal of BPHs feeding behavior was stored in the computer by amplification by an A/D converter (Di-710; Dataq Instruments, Akron, OH, USA) and then transformed into a waveform and displayed on the screen using the PROBE v3.4 software package (Wageningen Agricultural University, Wageningen, The Netherlands). Stylet+a software was used to distinguish and annotate manually the waveform events, such as np for nonpenetration, ph for pathwayphase (including N1 for penetration initiation, N2 for stylet movement and salivation, and N3 for extracellular activity near the phloem region), N4-a for intracellular activity in the phloem tissue, N4-b for sustained phloem sap ingestion in the phloem tissue, and N5 for stylets in the xylem tissue, termed according to Seo et al. [24] and He et al. [25]. The stylet activity was recorded continuously for 6 h, and 13 effective repetitions were selected for statistical analysis. Experiments were conducted inside a Faraday cage in the laboratory to shield it from external, electric noise at 25 ± 2 °C and 80% RH (relative humidity) conditions.

### 4.4. Transcriptome Resequencing, Gene Expression, and Real-Time PCR Analysis

#### 4.4.1. RNA Extraction

Total RNA was extracted from the third-instar nymphs of BPHs that fed on the rice sprayed with 100 dilution multiple concentration of GC film-forming emulsion for 36 h, using RNAiso Plus (Takara). Control RNA was extracted from the BPHs that fed on the rice sprayed with distilled water. The RNA extracted from the BPHs that fed on the rice sprayed with distilled water was used as control. RNA concentration, and purity was measured using NanoDrop 2000 (Thermo Fisher Scientific, Wilmington, DE, USA), while RNA integrity was assessed using the RNA Nano 6000 Assay Kit of the Agilent Bioanalyzer 2100 system (Agilent Technologies, Santa Clara, CA, USA).

#### 4.4.2. Library Construction and Sequencing

A total amount of 1 μg RNA per sample was used as input material for the RNA sample preparations. Extracted RNA was transcribed into cDNA, and sequencing libraries were generated using NEBNext UltraTM RNA Library Prep Kit for Illumina (NEB, USA), then sequenced on an Illumina platform, and paired-end reads were generated after cluster generation.

#### 4.4.3. Illumina Reads Processing and Assembly

The quantification of the RNA (including lncRNA and mRNA) transcriptome was performed using hisat2. The parameters were defaulted. The alignment rates were as follows: Q10, which corresponds to a base accuracy of 90%, Q20, which corresponds to a base accuracy of 99%, Q30, which corresponds to a base accuracy of 99.9%, and Q40, which corresponds to a base accuracy of 99.99%. Raw data (raw reads) of fastq format were firstly processed through in-house perlscripts. In this step, clean data (clean reads) were obtained by removing reads containing adapter, reads containing ploy-N, and low-quality reads from raw data. All the downstream analyses were based on clean data with high quality.

#### 4.4.4. Differential Expression Genes Analysis

The level of gene expression was quantified by FPKM (fragments per kilobase of transcript per million mapped reads) [71]. Differentially expressed genes (DEGs) analysis was carried out for each sample with respect to control and between samples of BPHs that fed on the rice sprayed with GC film-forming emulsion. The DEGs analysis included three steps: read counts normalization, model-dependent *p*-value estimation, and FDR (false discovery rate) value estimation were performed using the DESeq R package (1.18.0) based on multiple hypothesis testing. FDR (adjusted *p*-value) value <0.05 and |log2 (fold change)| >1.0 were set as the threshold for significantly differential expression. Differential expression analysis was performed by the likelihood ratio test method of the edgeR package [72], and the results were visualized as volcano plots and heat maps. The expression level of DEGs was higher in treated than in control, which was considered to be upregulated, whereas those exhibiting the opposite relationship were considered to be downregulated.

#### 4.4.5. Annotation of Unigenes

The comparative analysis involved gene functional annotation (such as Nt—NCBI nonredundant nucleotide sequences, Pfam—protein family, KOG/COG—Clusters of Orthologous Groups of proteins, Swiss-Prot—a manually annotated and reviewed protein sequence database, KO—KEGG Ortholog database, and GO—Gene Ontology), differential expression analysis, GO enrichment analysis, KEGG pathway enrichment analysis, followed the methods described by References [72,73,74].

#### 4.4.6. Repellency-Related Genes Analysis

Referring to the method of Xu et al. [57], sequences of the BPH chemosensory genes in DEGs from our transcriptome were obtained by searching the GenBank with the keywords “odorant-binding proteins genes (OBPs), chemosensory proteins genes (CSPs), odorant receptors genes (ORs), ionotropic receptors genes (IRs), gustatory receptors genes (GRs), and sensory neuron membrane proteins genes (SNMPs), and so on”. The sequences of BPH chemosensory genes were searched, and sequence alignment was performed using the program TBLASTN (version 2.7.1).

#### 4.4.7. Validation by Quantitative Real-Time PCR

Validation of repellency-related genes was performed using quantitative real-time PCR (qRT-PCR). Total RNA was extracted from tissues by Trizol (Invitrogen, Waltham, MA, USA). The extracted RNA concentration was measured using NanoDrop ND-2000 Spectrophotometer (scandrop100), and the purity of the samples was determined by the OD ratios, A260/A280, then samples were reverse-transcribed into cDNA in the following reaction procedures: 37 °C for 60 min, 85 °C for 5 min, and 4 °C for 5 min. cDNA was amplified by Aidlab reverse transcription PCR kit (TUREscript 1st Stand cDNA SYNTHESIS Kit) through the following procedures: 40 cycles of 95 °C for 3 min, 95 °C for 10 s, 60 °C for 30 s, 95 °C for 10 s, and 60 °C for 30 s, melt curve analysis (60 °C~95 °C, +1 °C/cycle, holding time 4 s). The relative expression of the target gene was calculated by using 2^−ΔΔCt^ method with *U6* as the internal reference gene.

### 4.5. Statistics Analysis

The data are presented as the mean ± standard deviation and were tested for statistical differences using a one-way analysis of variance (ANOVA) followed by post hoc least significance difference (LSD) by SPSS 17.0 software (SPSS, Chicago, IL, USA). Transcriptomic sequence analysis was conducted by NCBI (https://www.ncbi.nlm.nih.gov/, accessed on 9 October 2021). Differential expression analysis was performed by the likelihood ratio test method of the edge R package (Robinson et al., 2010). Finally, the results were visualized as volcano plots and heat maps. Functional enrichment analysis was performed using the cluster Profiler package, including GO and KEGG pathway [73].

## 5. Conclusions

To summarize, the GC film had a repellency effect against BPHs, with 60.00% repellency in repellent activity test and 73.93% in olfactory reaction analysis. GC film repellency against BPHs relies on smell, by olfactory reaction analysis. EPG analysis showed that np waveforms were significantly influenced after treatment with GC film, which indicated that GC film affected recognition of BPHs to rice. Transcriptome analysis showed GC film repellency against BPHs is a result of inhibition of the expression of OBP genes and simultaneous promotion of the expression of GR/OR genes. *OBPs gene 797* and *GRs/ORs*
*gene 13110* appeared to be the crucial candidate genes for GC film repellency against BPHs. These results provide the first demonstration of the potential use of GC film as a repellent against BPHs, and provide useful information for preliminarily understanding of the mechanism at physiological and molecular levels, which also could help in the development and improvement of GC films and may offer an alternative approach for the control of BPHs in the near future.

## Figures and Tables

**Figure 1 ijms-23-00758-f001:**
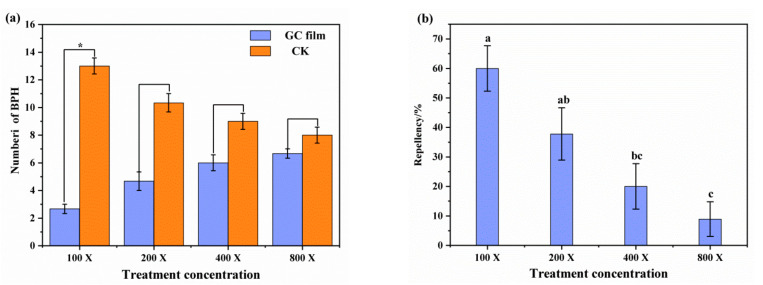
The feeding choice of BPHs and repellent effect of GC film on BPHs after treatment for 4 h. (**a**) The number of BPHs landed at GC film-treated rice seedlings (blue) and water-treated rice seedlings (brown). (**b**) The repellency of GC film to BPH at 100 dilution multiple (100×), 200 dilution multiple (200×), 400 dilution multiple (400×), and 800 dilution multiple (800×). The data are expressed as mean ± SD from three replicates. * different lowercase letters indicate significant difference between the control and treatment at *p* < 0.05. The same below.

**Figure 2 ijms-23-00758-f002:**
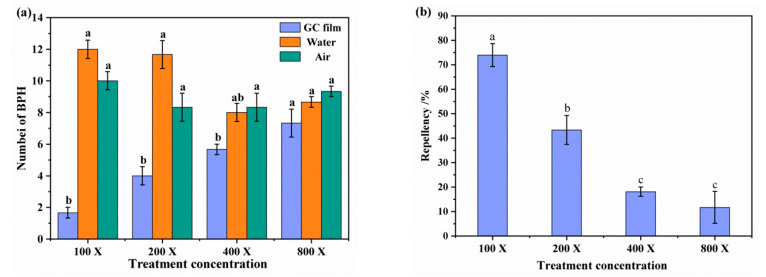
The olfactory reaction and repellent effect of GC film on BPHs after treatment for 4 h. (**a**) The number of BPHs in GC film (blue), water (brown), and air (green) flavor source treatment. (**b**) The repellency in different GC film treatments of 100 dilution multiple (100×), 200 dilution multiple (200×), 400 dilution multiple (400×), and 800 dilution multiple (800×). Water was used as control. The data are expressed as mean ± SD from three replicates.

**Figure 3 ijms-23-00758-f003:**
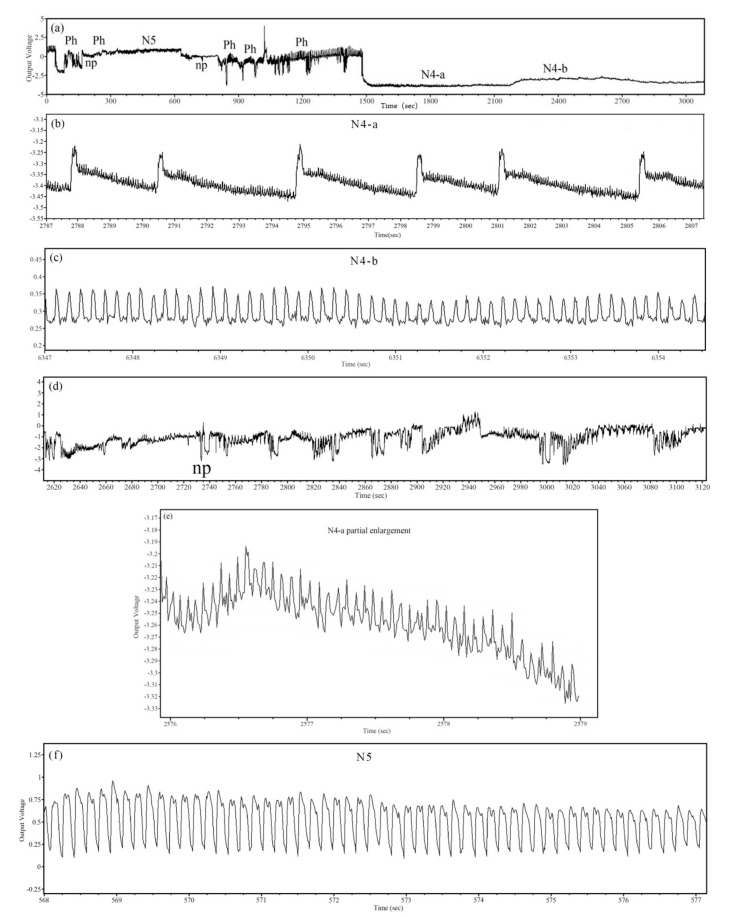
Analysis of EPG waveform characteristics of BPHs aspiration behavior. Integrity waveforms of 6 h EPG identified from BPH (**a**), local amplification feature of N4-a (**b**), N4-b (an intracellular activity of the scalpel within the phloem) (**c**), in detail for np (nonpenetration) (**d**), partial enlarged detail of N4-a (sustained phloem sap ingestion in the phloem tissue) (**e**), N5 (stylets in the xylem tissue termed putative xylem ingestion phase) (**f**).

**Figure 4 ijms-23-00758-f004:**
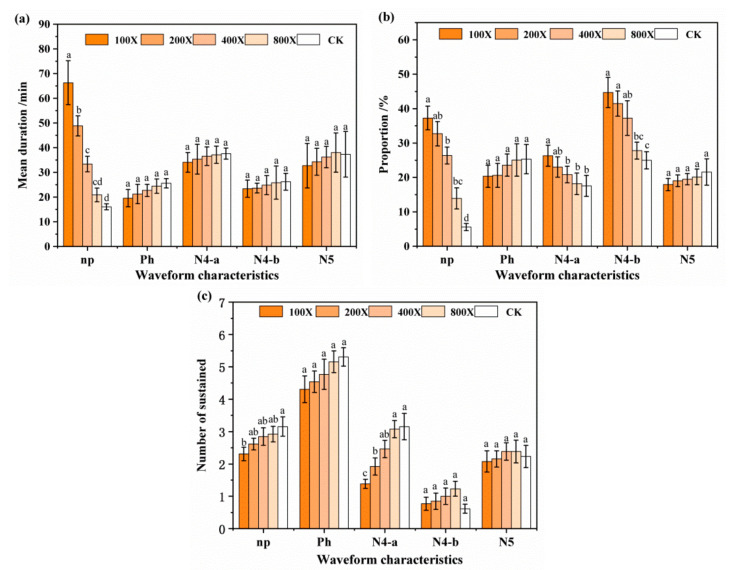
Comparison of EPG response variables of BPH feeding on rice plants after treatment with different dilution multiples of film-forming emulsion. (**a**) Mean duration, (**b**) proportion, (**c**) number of sustained np, Ph, N4-a, N4-b, and N5 EPG waveforms in different GC film treatments of 100 dilution multiple (100×), 200 dilution multiple (200×), 400 dilution multiple (400×), and 800 dilution multiple (800×), CK—used water as control. All experiments were repeated thirteen times, and the data are presented as mean ± SD.

**Figure 5 ijms-23-00758-f005:**
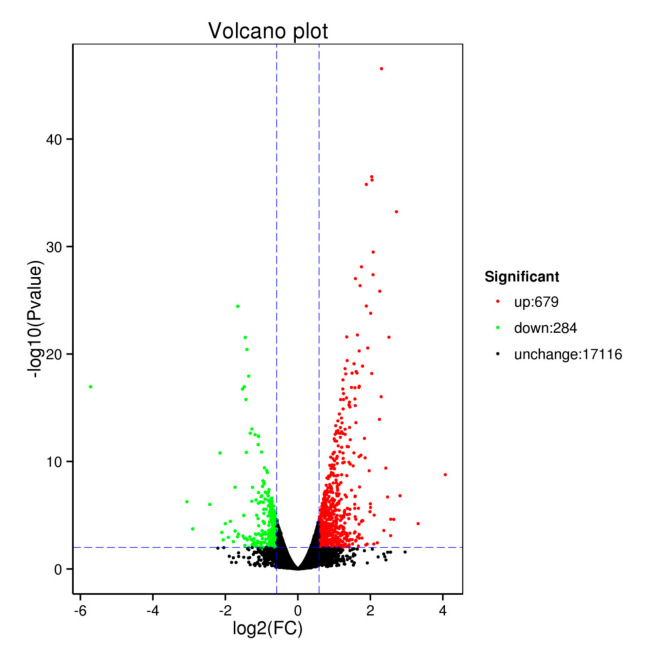
Volcanic map of differentially expressed genes.

**Figure 6 ijms-23-00758-f006:**
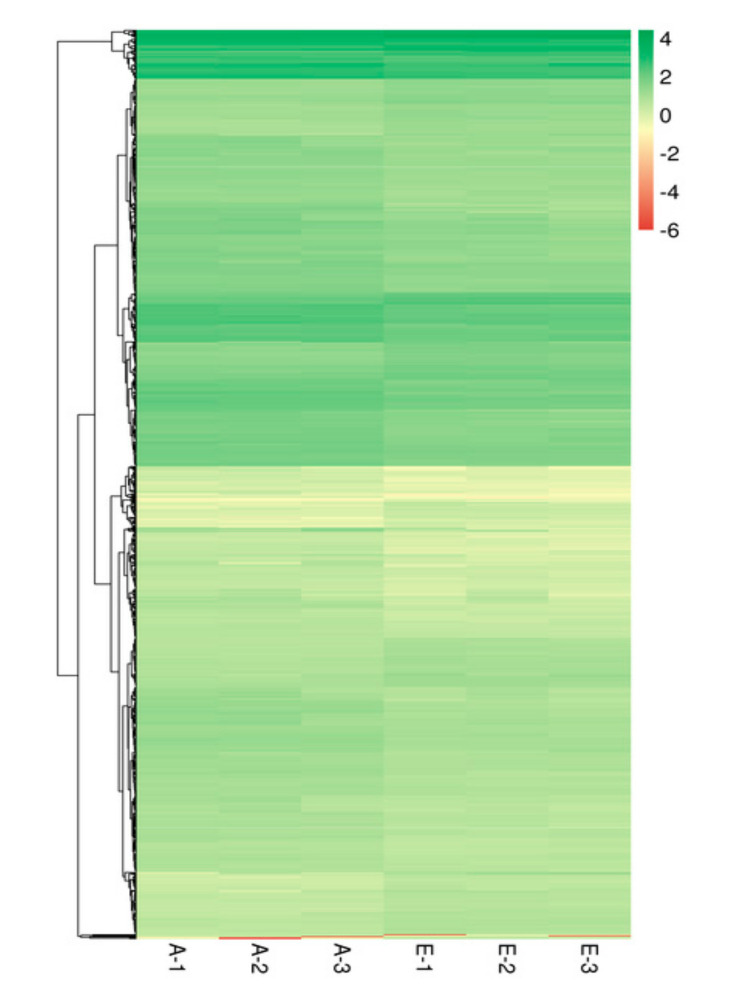
The cluster analysis of differentially expressed genes.

**Figure 7 ijms-23-00758-f007:**
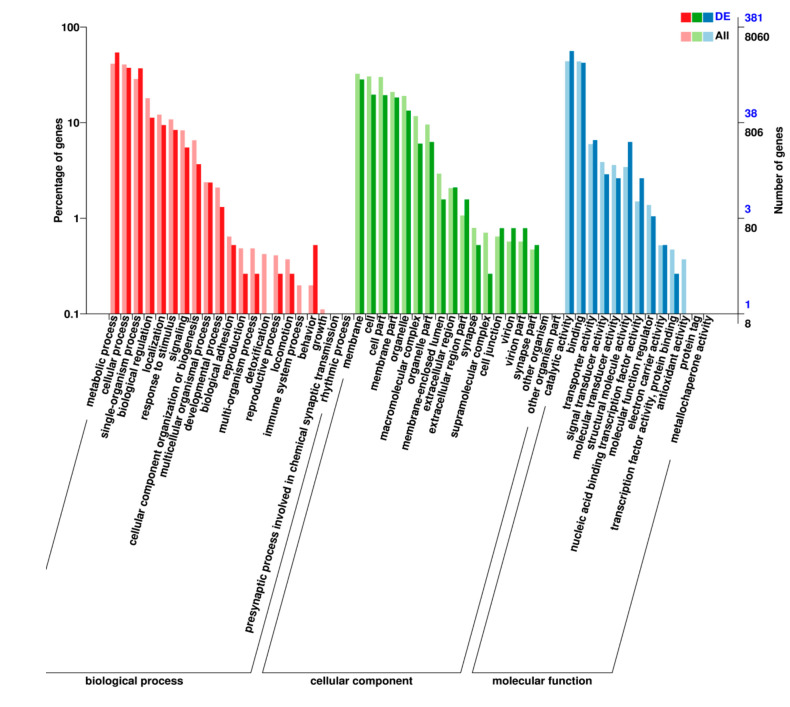
Gene Ontology (GO) statistical histogram of differentially expressed genes.

**Figure 8 ijms-23-00758-f008:**
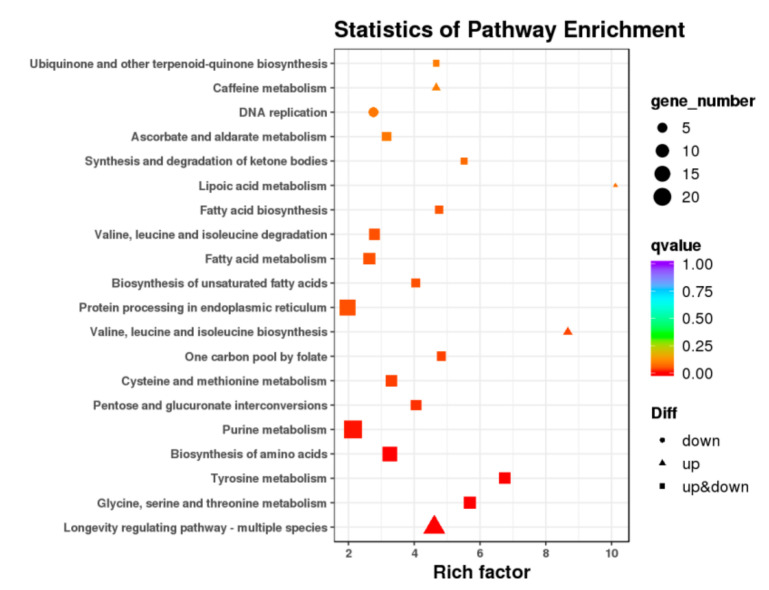
The KEGG enrichment pathways of differentially expressed genes.

**Figure 9 ijms-23-00758-f009:**
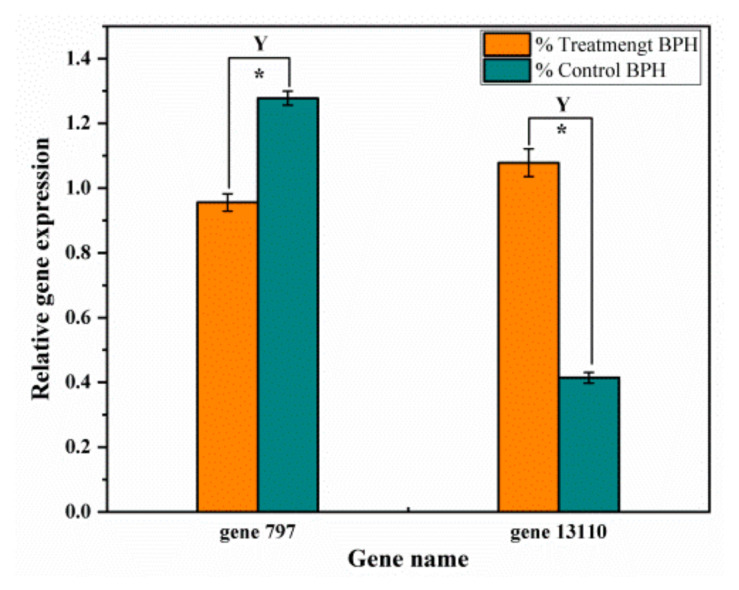
Relative expression levels of two genes (*gene 797* and *gene 13110*) in BPHs treatment and control groups validated through qRT-PCR analysis. *—indicated significant difference between treatment and control of relative gene expression (*p* < 0.05), Y—indicated that the up/downregulated of qRT-PCR was consistent with the transcriptome analysis.

**Figure 10 ijms-23-00758-f010:**
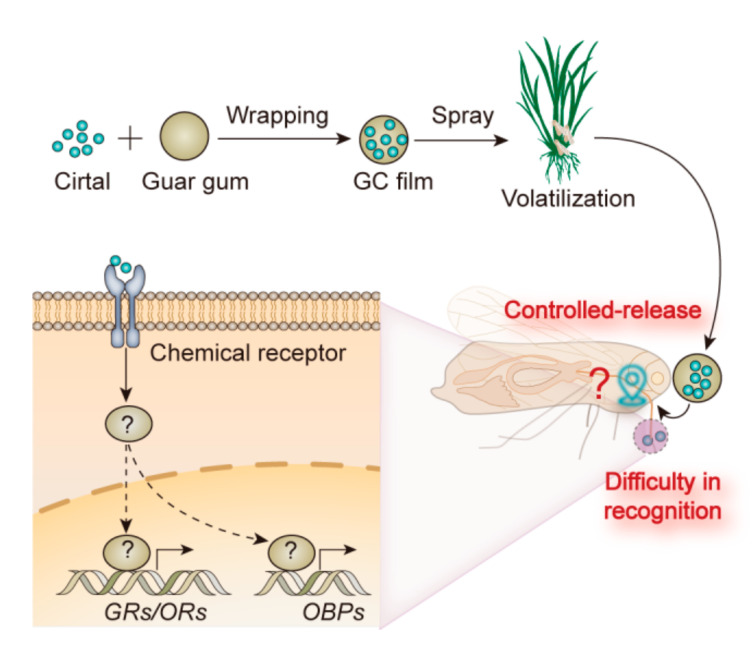
Schematic plot for repellency mechanism of GC film on BPHs. The citral was wrapped by guar gum and made into GC film. Then, rice was sprayed with film-forming emulsion of GC film. By the way of controlled release of citral, GC film influenced the recognition of BPHs to rice. Further, the citral, which is the main active constituent of GC film, affected expression of OBPs gene and GRs/ORs gene levels, resulting in interference in the recognition and localization of BPHs.

**Table 1 ijms-23-00758-t001:** Summary of transcriptome sequencing-related information of BPH after 48 h feeding on GC film-forming emulsion-treated rice.

Samples	No. of Clean Sequence	No. of Clean Base (bp)	GC Percent (%)	≥Q30 Percent (%)
A1	26,374,790	7,884,476,620	47.16%	93.63%
A2	24,581,687	7,348,758,594	47.12%	94.02%
A3	23,720,236	7,091,346,904	47.38%	93.55%
E1	28,026,245	8,380,871,624	46.88%	94.16%
E2	21,571,083	6,448,578,180	46.87%	94.21%
E3	26,028,546	7,778,099,976	47.58%	93.63%

**Table 2 ijms-23-00758-t002:** Differential expression analysis of chemosensory system genes in BPHs.

Gene Categories	Gene Name	Gene Description	Differential Expression Multiple	Up- or Downregulated	Significance
OBPs	*gene 797*	odorant binding	0.66	down	yes
GRs/ORs	*gene 13110*	gustatory and odorant receptor	2.55	up	yes

**Table 3 ijms-23-00758-t003:** Gene testing information: the selective genes and their primers used in qRT-PCR.

Gene Categories	Gene Name	Forward Primer (5′-3′)	Reverse Primer (3′-5′)	Amplicon Length
OBPs	*gene 797*	TTCATTGCCTGTGGATTCAACTC	CACGCTTGTCCTGTTCATCATAG	109 bp
GRs/ORs	*gene 13110*	CTTGCGTGCGAGAGTGGA	AACCTGCGACGAGTTGCT	77 bp
Reference gene	*U6*	CTCGCTTCGGCAGCACA	AACGCTTCACGAATTTGCGT	107 bp

## Data Availability

Data supporting reported results can be found at Appendix A.

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
