# Peer review of "Repellency Mechanism of Natural Guar Gum-Based Film Incorporated with Citral against Brown Planthopper, Nilaparvata lugens (Stål) (Hemiptera: Delphacidae)"

_ijms, 2022, doi:10.3390/ijms23020758_

Round 1
Reviewer 1 Report
The authors present the results of an exciting investigation where citral is used as a repellent agent dissolved in guar gum against Nilaparvata lugens. The Authors present two types of results, behavioral reactions of insects on citral formula, secondary, changes in odorant genes expression profile. The presented formula of citral is an interesting and intriguing solution. The project is well planned and carried out, while its description in a manuscript requires extensive correction and development. I have many serious considerations, which I listed below:
The fundamental problem, in my opinion, is an incorrectly selected control group, the authors use plants sprayed with water, and it should be a clean carrier without an additive. Why was water used and not a guar gum without citral?
The article undoubtedly requires a very advanced language correction due to numerous errors (I do not mark them in the text, there are too many of them)
The authors should explain in M&M why only females were used in the experiments
Captions under figures should refer to specific graphs in the graphic and explain the letters assigned to them. Many descriptions lack information about statistics.
In the results, the authors should use a more advanced analysis than a simple comparison of the following types of signals for EPG, e.g. PCA, decomposing data that depend on each other does not make sense, if the registration lasts a specific time, it is known that when time of one wave increases, the time of other wave decreases. These data are bound together.
The abbreviations describing the subsequent phases/waves of EPG should have consistent formatting, e.g., NP nor np. At times it is not easy to understand
The discussion is a simple repetition of the results and does not refer to the obtained results at the level of inference, nor does it analyze them in the context of other articles (repellent citral action, changes in gene expression, behavioral reactions). It requires a thorough redrafting and adding threads.
Author Response
Thanks to the reviewers for their constructive suggestions, our point-by-point response to the reviewer’s comments was done. Please see the attachment. Thanks very much for your detailed comments again!

Reviewer 2 Report
The paper is not written clearly. The results are presented in an interesting way.
All the "?" in the text should be explained.

Author Response

(The authors gave the same response as above.)

Round 2
Reviewer 1 Report
Dear Authors, thank you for all modifications. Please check the language.
Author Response
Thanks for your positive comments on our study and we will polish the manuscript further. your comments are all valuable and very helpful for revising and improving our manuscript, as well as the important guiding significance to our researches.
There are really some language mistakes in our manuscript. Author would like to thank constructive comments of reviewer and had amended it. I have marked text in blue. Futher, our paper will be polished by professional institute. Thedetailed explanation were on attachment. Please see the attachment.
